# Associations between dietary microbe intake and mortality risk in individuals with sleep disorders: Evidence from NHANES

Xuanchun Huang[1‡], Lanshuo Hu[2‡], Ruikang Liu[1‡], Jun Li[1]*, Tiantian Xue[1]*

1 Guang'anmen Hospital, China Academy of Traditional Chinese Medicine, Beijing, China, 2 Xiyuan Hospital, China Academy of Traditional Chinese Medicine, Beijing, China

‡ Xuanchun Huang, Lanshuo Hu and Ruikang Liu contributed equally to this work and share first authorship
* gamyylj@163.com (JL); 15776760213@163.com (TX)

## Abstract

### Objective

To investigate the association between dietary microbial intake, sleep patterns, and all-cause and cardiovascular mortality among U.S. adults.

### Methods

This study is conducted using data from the 2005–2014 National Health and Nutrition Examination Survey (NHANES). Kaplan-Meier curves are used to preliminarily explore the relationship between dietary microbial intake, sleep disorders, and all-cause and cardiovascular mortality in the population. The Cox proportional hazards model is applied for both individual and combined analyses to examine the relationship between dietary microbial intake, sleep disorders, and mortality risk, with subgroup and sensitivity analyses performed to assess model stability.

### Results

This study included 21,233 participants, among whom 2,814 all-cause deaths and 877 cardiovascular deaths were documented. Kaplan-Meier survival analysis revealed a significant association between low dietary microbial intake or sleep disorders and elevated mortality. Cox proportional hazards modeling showed that, among individuals with sleep disorders, those with moderate dietary microbe intake had a lower mortality hazard ratio compared to those with low intake. Conversely, the combination of low dietary microbe intake and sleep disorders was associated with the highest all-cause and cardiovascular mortality. Subgroup and sensitivity analyses demonstrated consistent associations across prespecified strata, with the inverse relationship between dietary live microbe intake and sleep disorder–related mortality remaining robust after adjustment for confounders.

**Data availability statement:** All relevant data are within the paper and its Supporting information files.

**Funding:** This study was supported by the National Natural Science Foundation of China (No. 82474494), National Key Research and Development Program of China (No. 2022YFC3500102), the Beijing Municipal Science and Technology Development Funding Program of Traditional Chinese Medicine (No. JJ-2020-69), and High Level Chinese Medical Hospital Promotion Project (No. HLCMHPP2023065). The funders had no role in study design, data collection and analysis, decision to publish, or preparation of the manuscript.

**Competing interests:** The authors have declared that no competing interests exist.

**Abbreviation:** NCHS, National Center for Health Statistics; BMI, Body Mass Index; PIR, Price-to-Income Ratio; NHANES, National Health and Nutrition Examination Survey; CVD-cause, Cardiovascular disease-caused mortality; ALL-cause, All reasons-caused mortality; HR, Hazard Ratios; CI, Confidence Intervals.

## Conclusion

Low dietary microbial intake and sleep disorders were independently and jointly associated with higher rates of all-cause and cardiovascular mortality in population. The observed inverse association between higher dietary microbial intake and mortality outcomes, particularly among individuals with sleep disorders, suggests a potential protective trend.

## 1. Introduction

The complex interplay between human gut microbiota and health has emerged as a frontier in scientific inquiry. Dietary live microorganisms, as exogenously sourced dietary components [1], play a pivotal role in shaping gut microbiota composition [2], while profoundly influencing metabolic health and immune functions [3,4], which has generated considerable interest in the daily intake of dietary microorganisms within clinical nutrition and preventive medicine fields. Although the term lacks precision, dietary live microorganisms are broadly understood to encompass active microbes capable of surviving the gastrointestinal environment and exerting physiological benefits post-ingestion [5]. These include, though are not limited to, strains such as Lactobacillus, Bifidobacterium, and Saccharomyces [6–8]. By modulating gut microbial balance [9], regulating gut-derived neurotransmitters [10], and fostering beneficial metabolic byproducts [11], these organisms may mitigate the risk of chronic diseases like hypertension, diabetes, and hyperlipidemia [12–14], and alleviate conditions related to mental health, such as anxiety, depression, and sleep disorders [15,16].

Epidemiological studies reveal that around 30% to 40% of adults experience varying degrees of sleep disturbances, including insomnia, sleep apnea, or circadian misalignment, often co-occurring with metabolic and psychological conditions [17–20]. Such sleep disorders are increasingly recognized as critical detriments to overall health. Consequently, identifying lifestyle interventions to offset the adverse impacts of poor sleep on health has become imperative. Recent studies suggest a potential link between gut microbiota and sleep regulation, likely mediated by various neuroimmune and endocrine pathways, including the gut-brain axis, which influences circadian rhythms [21]. Specifically, gut microbiota may produce bioactive substances like short-chain fatty acids [22], bile acids [23], and a spectrum of neurotransmitters [24] that can directly or indirectly modulate brain function, thereby impacting sleep quality or even inducing sleep disorders. Moreover, sleep deprivation itself may exacerbate eating behaviors through mechanisms such as chronic inflammation, endocrine disruption, and autonomic dysfunction [25–27], contributing to gut microbial imbalance and elevating the risk of cardiovascular disease and mortality [28,29]. Notably, recent evidence suggests that timely intake of dietary microorganisms may improve sleep by rebalancing gut microbiota [30–32], although robust evidence remains limited on their potential to counteract the health effects of sleep disturbances.

In consideration of the aforementioned perspectives, this study seeks to determine whether dietary intake of live microbes can improve long-term health outcomes for

individuals with sleep disorders. Using data from the U.S. National Health and Nutrition Examination Survey (NHANES), we will investigate the associations between dietary microbial intake, sleep patterns, and mortality related to all causes and cardiovascular conditions. By clarifying these relationships, this study aims to contribute novel insights into cardiovascular prevention strategies for those with sleep disorders, advancing a more integrated understanding of population health and chronic disease management.

## 2. Materials and methodology

This study utilized the NHANES database, conducted by the National Center for Health Statistics (NCHS). NHANES is a comprehensive survey designed to gather nationally representative data on the health and nutrition of the U.S. civilian population, encompassing demographic, socioeconomic, dietary, and health-related information. To ensure sample diversity, NHANES employs a stratified, multistage sampling approach to select participants from across the nation. The study protocol received approval from the NCHS Research Ethics Review Board at the Centers for Disease Control and Prevention, with written informed consent obtained from all participants. Further details are available on the NHANES official website.

### 2.1. Inclusion and exclusion criteria for participants

This study gathered data from 50,965 participants published on the NHANES website between 2005 and 2014, applying the following exclusion criteria: (1) individuals under 20 years of age, (2) pregnant individuals, (3) individuals with missing dietary data, (4) individuals with missing sleep data, (5) individuals lacking survival data, and (6) individuals with missing key covariates. Ultimately, 21,233 participants were included in the study. Details are shown in Fig 1.

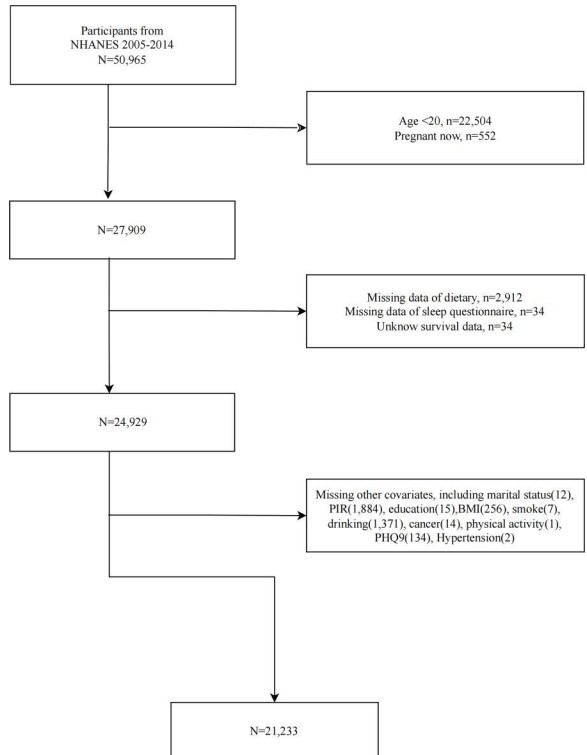

**Fig 1. Flow diagram illustrating the participant selection process.**

## 2.2. Assessment of dietary live microbes and sleep disorders

All dietary microbe intake data were collected through two 24-hour dietary recall interviews. The first dietary recall interview was conducted at the Mobile Examination Center (MEC), while the second interview was performed via telephone 3 to 10 days later. Participants were asked to recall all foods and beverages consumed during the previous 24-hour period (from midnight to midnight). To assist with accurate reporting, participants were provided with a set of measurement guidelines and a food model booklet to help describe food portions and details.

Previously, experts in the field evaluated 48 food subgroups within the NHANES database, analyzing the live microbial content of 9,388 food items. Consequently, the daily intake of dietary microbes is already known. Generally, foods that are sterile or pasteurized have very low levels of live microorganisms, while unpeeled raw fruits and vegetables contain moderate levels, and unprocessed fermented foods typically have high levels of live microorganisms. Accordingly, foods were classified into categories based on microorganism content: low ($10^4$ CFU/g), moderate ($10^4$–$10^7$ CFU/g), or high (>$10^7$ CFU/g) levels of dietary microorganisms. Participants were categorized based on their intake into three groups: low dietary live microorganism intake (all foods were in the low-level microorganism category), moderate dietary live microorganism intake (foods included moderate-level microorganisms but not high-levels), and high dietary live microorganism intake (foods included high-level microorganisms). This classification approach has been documented and applied in multiple studies [33–35].

Additionally, sleep disorder diagnosis was determined through a self-reported questionnaire, asking participants, "Have you ever been diagnosed with a sleep disorder by a doctor or other health professional?" These disorders include several conditions such as insomnia, sleep apnea, circadian rhythm sleep disorder, hypersomnia, and parasomnias, which are characterized by difficulties in falling asleep, poor sleep continuity, excessive sleep, or abnormal sleep behaviors.

## 2.3. Assessment of mortality

The NCHS linked NHANES data with NDI data using identifiers such as Social Security numbers and birth dates to obtain survival status information for participants, with follow-up through December 31, 2019. If no match was found in the NDI, the participant was assumed to be still alive. Causes of death for deceased participants were classified according to the International Classification of Diseases, 10th Revision (ICD-10). In this study, cardiovascular-related mortality encompassed conditions such as rheumatic heart disease, hypertensive heart disease, ischemic heart disease, pulmonary heart disease, cardiomyopathy, endocarditis, cerebral hemorrhage, and cerebral infarction, with corresponding ICD-10 codes I00-I09, I11, I13, I20-I51, and I60-I69.

## 2.4. The covariates included in this study

This study considered multiple variables that may influence the relationship between dietary live microorganisms, sleep disorders, and risks of cardiovascular and all-cause mortality. These variables covered various demographic characteristics of the study population, including age, gender, race, education level, income-to-poverty ratio, and BMI, as well as lifestyle factors such as smoking and drinking habits, hypertension, diabetes, hyperlipidemia, physical activity, presence of depression, and daily caloric intake. Among them, smoking history was defined as having smoked more than 100 cigarettes in one's lifetime, while drinking status was defined as at least 12 drinks annually. One drink unit was defined as equivalent to 12 ounces of beer, 5 ounces of wine, or 1.5 ounces of distilled spirits. Hypertension, diabetes, and hyperlipidemia were identified based on physician diagnoses, lab tests, or use of related medications. Depression was assessed using the PHQ-9 scale, with a score above 10 indicating a depressive state. Physical activity levels were categorized based on energy expenditure in daily work and recreational activities. Detailed information is available on the NHANES website.

## 2.5. Statistical analyses

Statistical analyses in this study were rigorously conducted following the design methods recommended by the NHANES database, with appropriate weights applied to each analysis. For continuous variables following a normal distribution, data were presented as mean ± standard deviation; for those not following a normal distribution, the median was reported. Baseline differences for continuous variables were assessed using analysis of variance (ANOVA), while categorical variables were evaluated using the $\chi^2$ test, with results expressed as percentages.

The Kaplan-Meier method was used to initially assess the relationship between dietary live microorganisms, sleep disorders, and all-cause and cardiovascular mortality. Subsequently, a Cox proportional hazards regression model was applied to explore the impact of dietary live microorganisms and sleep disorders on all-cause and cardiovascular mortality. Cox regression was also conducted for subgroup analyses based on different levels of dietary live microorganism intake and sleep status, with results expressed as hazard ratios (HR) and 95% confidence intervals (CI). Three models were used in the analysis: Model 1, unadjusted for confounders; Model 2, adjusted for age, gender, race, education level, PIR, and marital status to account for demographic influences; and Model 3, further adjusted for BMI, alcohol consumption, smoking, diabetes, hypertension, physical activity, energy intake, hyperlipidemia, cancer, and depression. Additionally, subgroup analyses were conducted to assess the stability and reliability of our findings. All analyses were performed using R software (version 4.3.1), with statistical significance set at a two-sided $P < 0.05$.

## 3. Results

### 3.1. Demographic and clinical attributes of participants

Based on the inclusion criteria, this study ultimately included 21,233 participants from NHANES 2005–2014 data, comprising 10,700 men and 10,533 women, with 1,836 patients diagnosed with sleep disorders, 2,814 cases of all-cause mortality, and 877 cardiovascular-related deaths. For analysis, participants were divided into three groups according to the amount of dietary live microorganisms. It was observed that individuals with higher dietary live microorganism intake tended to have higher incomes, lower BMI, higher levels of education, and a lower risk of chronic conditions such as diabetes, hypertension, hyperlipidemia, and depression. Additionally, those with higher dietary live microorganism intake had lower risks of cardiovascular and all-cause mortality compared to those in the low and moderate intake groups (see Table 1 for details).

Furthermore, histograms of participants across different years show that, regardless of sleep disorder status, individuals tended to follow a moderate dietary live microorganism intake. However, over time, the number of individuals opting for a high dietary live microorganism intake gradually increased. Concurrently, the prevalence of sleep disorders among participants also increased with each follow-up year (see Fig 2 for details).

### 3.2. The Kaplan-Meier curves between dietary live microbes and sleep disorders with mortality

To investigate the clinical significance of dietary live microorganisms and sleep status in population prognosis, this study used Kaplan-Meier analysis to preliminarily assess the associations between dietary live microorganism intake, sleep status, and all-cause and cardiovascular mortality. The results revealed a significant correlation between the amount of dietary live microorganisms and both all-cause and cardiovascular mortality. Individuals with high dietary live microorganism intake exhibited lower risks of ALL-cause ($P_{ALL} < 0.001$) and cardiovascular mortality ($P_{CVD} < 0.001$), while those with low intake had higher mortality risks.

Additionally, sleep status also impacted long-term prognosis: individuals with sleep disorders had a higher risk of all-cause mortality than those without sleep disorders ($P_{ALL} < 0.001$) and a similarly elevated risk of cardiovascular mortality ($P_{CVD} < 0.001$). The detailed results are presented in Fig 3.

**Table 1. Demographic and clinical characteristics of participants by dietary live microbes.**

| | total | Low | Middle | High | *P* |
|---|---|---|---|---|---|
| **n** | **21233** | **7589** | **8927** | **4717** | |
| **Age, year** | 47.121±0.289 | 45.216±0.336 | 48.773±0.335 | 46.904±0.418 | < 0.0001 |
| **PIR** | 3.032±0.039 | 2.628±0.048 | 3.124±0.035 | 3.390±0.049 | < 0.0001 |
| **BMI** | 28.863±0.086 | 29.364±0.114 | 28.731±0.097 | 28.446±0.156 | < 0.0001 |
| **energy intake, kcal/day** | 2197.829±10.705 | 2119.088±14.271 | 2188.927±16.031 | 2310.217±18.100 | < 0.0001 |
| **Sex** | | | | | < 0.0001 |
| male | 10700(49.487) | 4087(53.638) | 4449(48.418) | 2164(45.981) | |
| female | 10533(50.513) | 3502(46.362) | 4478(51.582) | 2553(54.019) | |
| **Race** | | | | | < 0.0001 |
| Non-Hispanic Black | 4465(10.618) | 2196(15.991) | 1637(9.236) | 632(6.078) | |
| Non-Hispanic White | 10174(71.038) | 3221(64.934) | 4261(71.071) | 2692(78.613) | |
| Other Race | 6594 (18.345) | 2172 (19.075) | 3029 (19.695) | 1393 (15.310) | |
| **Marital Status** | | | | | < 0.0001 |
| non-single | 12665(63.736) | 4136(58.839) | 5547(65.302) | 2982(67.389) | |
| single | 8568(36.264) | 3453(41.161) | 3380(34.698) | 1735(32.611) | |
| **Education** | | | | | < 0.0001 |
| Less than high school | 2013(4.932) | 807(5.972) | 945(5.559) | 261(2.648) | |
| high school | 8078(34.244) | 3388(42.401) | 3290(32.599) | 1400(26.640) | |
| Some college or above | 11142(60.824) | 3394(51.627) | 4692(61.842) | 3056(70.713) | |
| **Drinking** | | | | | 0.007 |
| No | 2785(10.469) | 1050(11.372) | 1204(10.863) | 531(8.721) | |
| Yes | 18448(89.531) | 6539(88.628) | 7723(89.137) | 4186(91.279) | |
| **Smoke** | | | | | < 0.0001 |
| No | 11340(53.853) | 3762(49.066) | 4844(54.757) | 2734(58.410) | |
| Yes | 9893(46.147) | 3827(50.934) | 4083(45.243) | 1983(41.590) | |
| **Hypertension** | | | | | < 0.001 |
| No | 12241(62.501) | 4327(62.132) | 5004(60.971) | 2910(65.369) | |
| Yes | 8992(37.499) | 3262(37.868) | 3923(39.029) | 1807(34.631) | |
| **Diabetes** | | | | | < 0.0001 |
| No | 17443(86.743) | 6239(86.928) | 7189(85.356) | 4015(88.693) | |
| Yes | 3790(13.257) | 1350(13.072) | 1738(14.644) | 702(11.307) | |
| **Hyperlipidemia** | | | | | 0.01 |
| No | 6279(29.985) | 2306(30.214) | 2504(28.680) | 1469(31.751) | |
| Yes | 14954(70.015) | 5283(69.786) | 6423(71.320) | 3248(68.249) | |
| **Cancer** | | | | | < 0.0001 |
| No | 19191(90.251) | 6978(92.211) | 8007(89.749) | 4206(88.590) | |
| Yes | 2042(9.749) | 611(7.789) | 920(10.251) | 511(11.410) | |
| **Physical activity** | | | | | < 0.0001 |
| No | 11789(51.380) | 4302(52.084) | 4935(51.229) | 2552(50.739) | |
| Moderate | 4913(24.833) | 1588(22.149) | 2142(25.914) | 1183(26.488) | |
| Vigorous | 4531(23.786) | 1699(25.767) | 1850(22.857) | 982(22.773) | |
| **Depression** | | | | | < 0.0001 |
| No | 19358(92.435) | 6709(89.433) | 8253(93.644) | 4396(94.285) | |
| Yes | 1875(7.565) | 880(10.567) | 674(6.356) | 321(5.715) | |
| **sleep disorder** | | | | | 0.826 |
| No | 19397(91.235) | 6909(91.138) | 8177(91.138) | 4311(91.509) | |

*(Continued)*

Table 1. (Continued)

| | total | Low | Middle | High | *P* |
|---|---|---|---|---|---|
| Yes | 1836(8.765) | 680(8.862) | 750(8.862) | 406(8.491) | |
| **ALL-cause death** | | | | | < 0.0001 |
| No | 18419(90.300) | 6479(89.094) | 7701(89.908) | 4239(92.424) | |
| Yes | 2814(9.700) | 1110(10.906) | 1226(10.092) | 478(7.576) | |
| **CVD-cause death** | | | | | 0.002 |
| No | 20356(97.133) | 7238(96.781) | 8547(96.923) | 4571(97.902) | |
| Yes | 877(2.867) | 351(3.219) | 380(3.077) | 146(2.098) | |

**Abbreviation:** BMI: Body Mass Index; PIR: Price-to-Income Ratio; CVD-cause: Cardiovascular disease-caused mortality; ALL-cause: All reasons-caused mortality.

### 3.3. Cox proportional hazards regression analysis of dietary live microbes and sleep disorders with mortality

To further explore the impact of dietary live microorganisms and sleep disorders on long-term population prognosis, we applied a Cox proportional hazards model to examine each factor separately. The results showed significant associations between dietary live microorganisms and both all-cause and cardiovascular mortality across the entire population ($P_{\text{All-middle}} < 0.0001$, $P_{\text{All-high}} < 0.0001$, $P_{\text{CVD-middle}} = 0.001$, $P_{\text{CVD-high}} = 0.001$). Similarly, sleep disorders were significantly associated with increased all-cause and cardiovascular mortality ($P_{\text{All-sleep}} < 0.0001$, $P_{\text{CVD-sleep}} = 0.038$).

Specifically, using low dietary live microorganism intake as a reference, both moderate and high intake were associated with lower all-cause mortality risk: $HR_{\text{All-middle}} = 0.747$ (0.680, 0.819), $HR_{\text{All-high}} = 0.779$ (0.688, 0.883). Consistent results were observed for cardiovascular mortality: $HR_{\text{CVD-middle}} = 0.725$ (0.595, 0.883), $HR_{\text{CVD-high}} = 0.710$ (0.548, 0.921). Similarly, compared to individuals without sleep disorders, those with sleep disorders had a higher risk of mortality: $HR_{\text{All-sleep}} = 1.341$ (1.165, 1.544) and $HR_{\text{CVD-sleep}} = 1.315$ (1.015, 1.702). Detailed results are provided in Tables 2 and 3.

### 3.4. Joint association of dietary live microbes and sleep disturbances with mortality

Given the significant individual impacts of dietary live microorganisms and sleep disorders on prognosis, this study aimed to analyze their combined effects. We hypothesized that individuals with no sleep disorders and a high intake of dietary live microorganisms would have the lowest mortality risk, using this group as the reference for analysis.

The results showed that among those without sleep disorders, individuals with low dietary microorganism intake had an increased risk of mortality: $HR_{\text{ALL-low \& sleep disorder (-)}} = 1.276$ (1.117, 1.457) and $HR_{\text{CVD-low \& sleep disorder (-)}} = 1.414$ (1.087, 1.839). Among those with sleep disorders, individuals with moderate dietary microorganism intake had a higher risk of all-cause mortality: $HR_{\text{ALL-middle \& sleep disorder (+)}} = 1.270$ (1.017, 1.587). Those with both low dietary microorganism intake and sleep disorders had the highest mortality risk: $HR_{\text{ALL-low \& sleep disorder (+)}} = 1.749$ (1.413, 2.163) and $HR_{\text{CVD-low \& sleep disorder (+)}} = 2.200$ (1.400, 3.457). In contrast, diets with moderate and high levels of dietary microbes are associated with lower mortality risks compared to low dietary microbe intake, suggesting that individuals with sleep disorders may benefit from increased dietary microbe supplementation. The remaining groups did not show statistically significant effects, so they are not highlighted in the discussion or results. Thus, individuals with sleep disorders and low dietary microorganism intake exhibited the highest risk of all-cause and cardiovascular mortality (see Tables 4 and 5).The effect relationships between each group and mortality risk are illustrated in Fig 4.

### 3.5. Subgroup analysis

To evaluate the model's stability across various populations, we conducted a subgroup analysis, focusing specifically on differences in dietary live microorganism intake among individuals with sleep disorders across different demographics. Key groups of interest included gender, diabetes, hypertension, and depression.

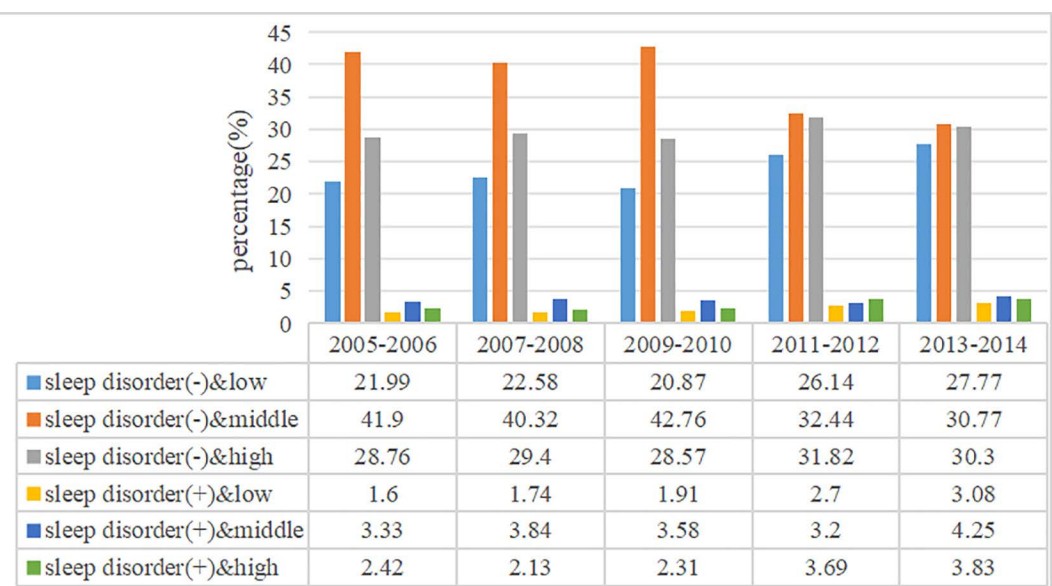

**Fig 2. Distribution of populations based on dietary microorganism intake and sleep disorder prevalence.**

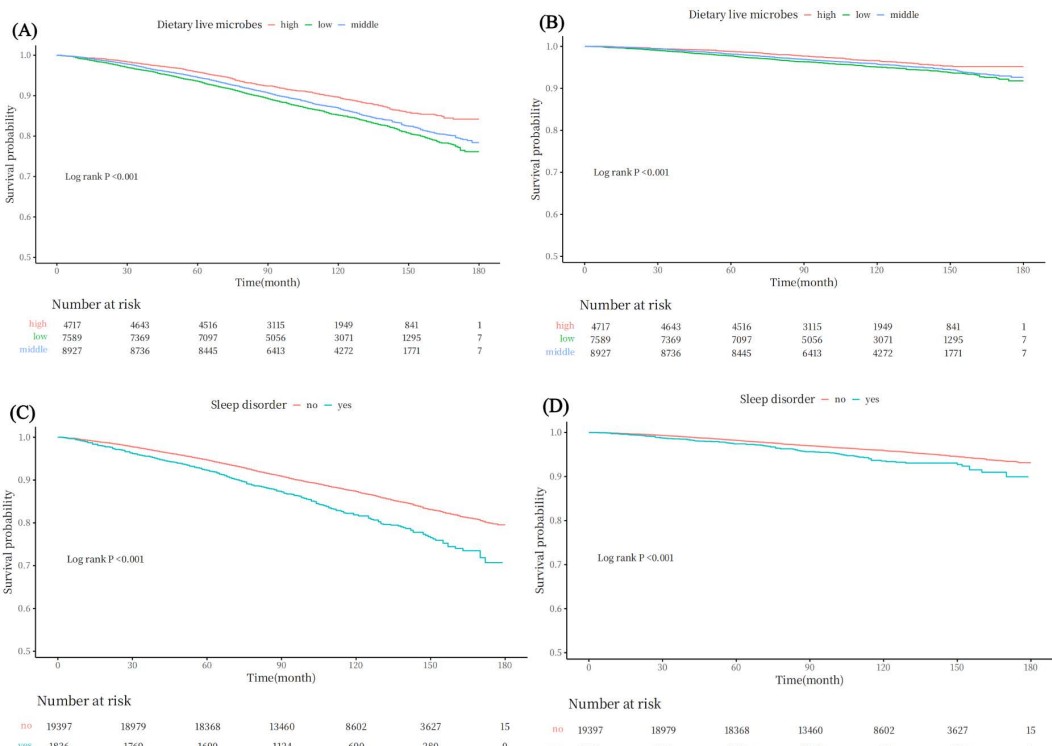

**Fig 3. Kaplan-Meier analysis of dietary microbes intake and sleep disorder impact on mortality: (A) Dietary microorganism intake and All-Cause mortality risk (B) Dietary microorganism intake and CVD-cause mortality risk (C) Sleep status and ALL-Cause mortality risk (D) Sleep status and CVD-cause mortality risk.**

**Table 2. Association Between dietary live microbes and sleep status with ALL-cause mortality across Cox proportional hazards models.**

| ALL-death | Dead/Alive | Model 1 | | Model 2 | | Model 3 | |
|---|---|---|---|---|---|---|---|
| | | HR (95%CI) | *P* | HR (95%CI) | *P* | HR (95%CI) | *P* |
| **Dietary live microbes** | | | | | | | |
| Low | 1110/7589 | ref | ref | ref | ref | ref | ref |
| Middle | 1226/8927 | 0.874(0.779,0.981) | 0.022 | 0.729(0.662,0.803) | <0.0001 | 0.747(0.680,0.819) | <0.0001 |
| High | 478/4717 | 0.698(0.614,0.794) | <0.0001 | 0.757(0.664,0.862) | <0.0001 | 0.779(0.688,0.883) | <0.0001 |
| **Sleep Disorder** | | | | | | | |
| No | 2496/19397 | ref | ref | ref | ref | ref | ref |
| Yes | 318/1836 | 1.628(1.431,1.854) | <0.0001 | 1.531(1.346,1.741) | <0.0001 | 1.341(1.165,1.544) | <0.0001 |

**Note:** Model 1: No adjustment for confounding factors; Model 2: Adjusted for age, sex, race, education level, PIR, and marital status to control for demographic influences; Model 3: Further adjusted from Model 2 to include BMI, alcohol consumption, smoking, diabetes, hypertension, physical activity, energy intake, hyperlipidemia, cancer, and depression.

**Table 3. Association Between dietary live microbes and sleep status with CVD-cause mortality across Cox proportional hazards models.**

| CVD-death | Dead/Alive | Model 1 | | Model 2 | | Model 3 | |
|---|---|---|---|---|---|---|---|
| | | HR (95%CI) | *P* | HR (95%CI) | *P* | HR (95%CI) | *P* |
| **Dietary live microbes** | | | | | | | |
| Low | 351/7589 | ref | ref | ref | ref | ref | ref |
| Middle | 380/8927 | 0.905(0.737,1.111) | 0.34 | 0.720(0.590,0.879) | 0.001 | 0.725(0.595,0.883) | 0.001 |
| High | 146/4717 | 0.654(0.504,0.848) | 0.001 | 0.702(0.542,0.910) | 0.007 | 0.710(0.548,0.921) | 0.001 |
| **Sleep Disorder** | | | | | | | |
| No | 778/19397 | ref | ref | ref | ref | ref | ref |
| Yes | 99/1836 | 1.494(1.161,1.923) | 0.002 | 1.486(1.157,1.909) | 0.002 | 1.315(1.015,1.702) | 0.038 |

**Note:** Model 1: No adjustment for confounding factors; Model 2: Adjusted for age, sex, race, education level, PIR, and marital status to control for demographic influences; Model 3: Further adjusted from Model 2 to include BMI, alcohol consumption, smoking, diabetes, hypertension, physical activity, energy intake, hyperlipidemia, cancer, and depression.

The findings revealed consistent trends across all-cause and cardiovascular mortality rates, with varying effect values across these populations, suggesting that our results remain stable across diverse groups. However, age, gender, diabetes, and depression demonstrated interactive effects on the association between dietary microorganism intake and sleep disorders with respect to all-cause mortality. In contrast, no interactive effects were found for cardiovascular mortality risk among those with sleep disorders. Further details on subgroup analyses regarding the long-term prognostic impact of dietary live microorganism intake and sleep disorders are provided in S1 and S2 Tables.

### 3.7. Sensitivity analysis

Considering the potential influence of social factors, processed food, dietary fiber intake, and related medications on dietary live microbe intake and the long-term prognosis of individuals with sleep disorders, this study conducted relevant sensitivity analyses by adjusting for key covariates such as social determinants of health (SDoH) [36,37] (including employment status, household income and poverty ratio, food security, education level, access to routine healthcare facilities, health insurance type, home ownership, and marital status), processed food intake, dietary fiber intake, and the use of antibiotics, hypnotics (sleeping pills), and probiotic supplements within the past 30 days. The results demonstrated that even after accounting for these covariates that may confound the association between dietary microbes and health outcomes in individuals with sleep disorders, those with both sleep disorders and low dietary microbe intake consistently

**Table 4. Association Between the Combined Effects of Dietary Live Microbes and Sleep Disorders on ALL-cause Mortality Across Cox Proportional Hazards Models.**

| Sleep Disorder | Dietary live microbes | ALL (Dead /Alive) | Model 1 | | Model 2 | | Model 3 | |
|---|---|---|---|---|---|---|---|---|
| | | | HR (95%CI) | P | HR (95%CI) | P | HR (95%CI) | P |
| No | High | 425/4311 | ref | ref | ref | ref | ref | ref |
| | Middle | 1085/8177 | 1.252(1.056,1.483) | 0.009 | 0.960(0.819,1.125) | 0.616 | 0.955(0.817,1.117) | 0.568 |
| | Low | 986/6909 | 1.428(1.247,1.635) | <0.0001 | 1.313(1.143,1.508) | <0.001 | 1.276(1.117,1.457) | <0.001 |
| Yes | High | 53/406 | 1.624(1.172,2.250) | 0.004 | 1.523(1.117,2.077) | 0.008 | 1.327(0.969,1.820) | 0.078 |
| | Middle | 141/750 | 2.015(1.626,2.497) | <0.0001 | 1.451(1.180,1.785) | <0.001 | 1.270(1.017,1.587) | 0.035 |
| | Low | 124/680 | 2.347(1.886,2.919) | <0.0001 | 2.040(1.664,2.499) | <0.0001 | 1.749(1.413,2.163) | <0.0001 |
| P for trend | | | <0.0001 | | <0.0001 | | <0.0001 | |

**Note:** Model 1: No adjustment for confounding factors; Model 2: Adjusted for age, sex, race, education level, PIR, and marital status to control for demographic influences; Model 3: Further adjusted from Model 2 to include BMI, alcohol consumption, smoking, diabetes, hypertension, physical activity, energy intake, hyperlipidemia, cancer, and depression.

**Table 5. Association Between the Combined Effects of Dietary Live Microbes and Sleep status on CVD-cause Mortality Across Cox Proportional Hazards Models.**

| Sleep Disorder | Dietary live microbes | CVD (Dead /Alive) | Model 1 | | Model 2 | | Model 3 | |
|---|---|---|---|---|---|---|---|---|
| | | | HR (95%CI) | P | HR (95%CI) | P | HR (95%CI) | P |
| No | High | 130/4311 | ref | ref | ref | ref | ref | ref |
| | Middle | 343/8177 | 1.455(1.153,1.837) | 0.002 | 1.077(0.855,1.355) | 0.529 | 1.076(0.852,1.360) | 0.538 |
| | Low | 305/6909 | 1.533(1.189,1.975) | <0.001 | 1.424(1.097,1.849) | 0.008 | 1.414(1.087,1.839) | 0.01 |
| Yes | High | 16/406 | 1.819(0.958,3.454) | 0.067 | 1.824(0.984,3.379) | 0.056 | 1.649(0.892,3.051) | 0.111 |
| | Middle | 37/750 | 1.672(1.029,2.719) | 0.038 | 1.221(0.773,1.928) | 0.392 | 1.077(0.674,1.721) | 0.757 |
| | Low | 46/680 | 2.688(1.687,4.283) | <0.0001 | 2.507(1.632,3.851) | <0.0001 | 2.200(1.400,3.457) | <0.001 |
| P for trend | | | <0.0001 | | <0.0001 | | <0.0001 | |

**Note:** Model 1: No adjustment for confounding factors; Model 2: Adjusted for age, sex, race, education level, PIR, and marital status to control for demographic influences; Model 3: Further adjusted from Model 2 to include BMI, alcohol consumption, smoking, diabetes, hypertension, physical activity, energy intake, hyperlipidemia, cancer, and depression.

exhibited the highest risks of all-cause mortality and cardiovascular mortality, highlighting the robust nature of the findings. Details are shown in Table 6.

## 4. Discussion

This cohort study included 21,233 participants from the NHANES database, surveyed between 2005 and 2014, to investigate the relationship between dietary live microorganism intake, sleep quality, and risks of all-cause and cardiovascular mortality. Results showed that, during follow-up, 877 participants died from cardiovascular causes and 2,814 died from various causes. At baseline, those with high dietary live microorganism intake exhibited higher survival rates. Preliminary Kaplan-Meier analysis revealed that high dietary live microorganism intake and healthy sleep patterns were associated with lower mortality rates. Further analysis through three Cox proportional hazards models indicated that moderate to high dietary live microorganism intake, compared to low intake, significantly reduced mortality risk, while sleep disorders elevated mortality risk. This led us to hypothesize a combined analysis of dietary microorganism intake and sleep disorders. Surprisingly, we found that individuals with sleep disorders and low dietary microorganism intake had a notably higher risk

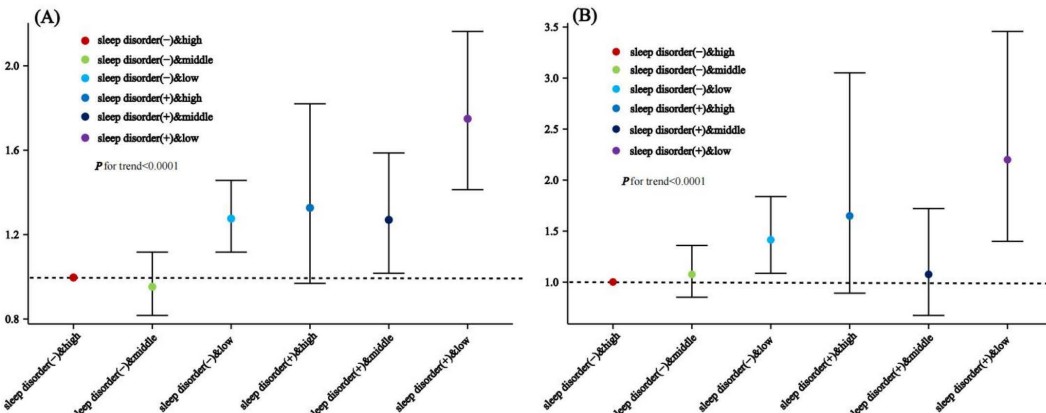

**Fig 4. Joint association of sleep status and Dietary live microbes with mortality: (A) Relationship between sleep status and dietary live microbes with ALL-cause mortality (B) Relationship between sleep status and dietary live microbes with CVD-cause mortality.**

**Table 6. Sensitivity analysis of the combined effects of dietary microbes and sleep status on long-term prognosis.**

| Sleep Disorder | Dietary live microbes | ALL-cause Mortality | | CVD-Cause Mortality | |
|---|---|---|---|---|---|
| | | HR (95%CI) | P | HR (95%CI) | P |
| No | High | ref | ref | ref | ref |
| | Middle | 0.967(0.824,1.135) | 0.684 | 1.087(0.856,1.381) | 0.495 |
| | Low | 1.239(1.085,1.414) | 0.002 | 1.405(1.086,1.818) | 0.010 |
| Yes | High | 1.368(0.992,1.887) | 0.056 | 1.676(0.901,3.120) | 0.103 |
| | Middle | 1.324(1.045,1.676) | 0.020 | 1.156(0.697,1.914) | 0.575 |
| | Low | 1.666(1.351,2.056) | <0.0001 | 2.230(1.400,3.552) | <0.001 |
| P for trend | | <0.001 | | <0.001 | |

Sensitivity Model: Further adjusted from Model 3 by including hypnotics, probiotic supplements, antibiotics, SDOH, Processed food intake, Dietary fiber intake.

of all-cause and cardiovascular mortality, while those with moderate to high microorganism intake showed a slight reduction in mortality risk. Our findings hold promise for guiding lifestyle improvements and dietary adjustments for individuals with sleep disorders.

Currently, research on the combined effects of dietary microorganism intake and sleep quality on mortality risk is limited. Most studies focus on the impact of probiotics on sleep or disease outcomes [38,39]. However, evidence related to probiotics indirectly supports our hypothesis, as dietary microorganisms inherently include a variety of probiotics [40]. Based on existing literature, we propose several potential mechanisms through which adequate dietary microorganism intake might improve long-term prognosis in individuals with sleep disorders.

Firstly, dietary microorganisms may mitigate cardiovascular risk factors associated with sleep disorders. Prolonged sleep disturbances disrupt hormone levels, including cortisol, leptin, and sex hormones, leading to metabolic imbalances, fat accumulation, and increased cardiovascular risk [41–43]. Dietary microorganisms could counteract these effects. For example, Larsen IS found in mouse experiments that probiotic diets prevented weight gain, enhanced insulin sensitivity, and reduced the risk of fatty liver [44]. Similarly, Pontes KSDS reported in a meta-analysis of 26 high-quality randomized controlled trials that probiotics reduced cardiovascular risk factors such as BMI, waist circumference, fat distribution, and LDL cholesterol [45]. Tiderencel KA's meta-analysis of nine clinical studies further demonstrated that probiotics improved

fasting glucose, plasma insulin, and glycated hemoglobin levels [46]. These findings suggest that probiotics in dietary microorganisms can attenuate cardiovascular risks induced by sleep disorders. Secondly, dietary microorganisms may improve sleep quality by regulating neurotransmitter levels, thereby alleviating the adverse effects of sleep disorders. According to Ho YT, daily supplementation with dietary microorganisms increased dopamine and serotonin levels in the bloodstream, enhancing sleep efficiency during deep sleep phases and reducing wakefulness [31]. Murack M reported that probiotics elevated tryptophan, serotonin, and L-lactate levels in the prefrontal cortex and hippocampus of mice, improving their sleep patterns [47]. Freitas SM found beneficial effects of probiotic-fermented milk on brainwave activity and sleep efficiency in rats [48], while Zheng Y reported that probiotic supplementation alleviated oxidative stress and inflammatory responses in the brains of sleep-restricted mice [49]. Although these findings are largely based on studies involving probiotics, they highlight how dietary microorganisms, through the gut-brain axis, can enhance sleep quality and mitigate adverse outcomes such as cardiovascular events, tumors, anxiety, depression, and immune dysfunction linked to sleep disorders [50–52]. In summary, dietary microorganism supplementation may help reduce the negative effects associated with sleep disorders, thereby extending lifespan and improving long-term prognosis.

Additionally, in our subgroup analysis, we observed that age, gender, diabetes, and depression interact with the relationship between dietary microorganism intake, sleep disorders, and all-cause mortality. This suggests that dietary microorganism intake and sleep disorders have a more pronounced impact on health outcomes in younger men compared to women and older adults. This may be due to the greater social pressures and unhealthy lifestyle habits often found in young adult men [53–56], meaning that correcting dietary habits and improving sleep in this group could yield substantial health benefits. Similarly, among individuals with diabetes and depression, the impact of dietary microorganism intake and sleep quality on health outcomes appears to be less significant than in the general population. Diabetic individuals face elevated risks of cardiovascular disease, cancer, and infections, which may lead to increased mortality [57,58]. For those with depression, suicide risk, as well as cancer and cardiovascular issues associated with depression, may contribute to higher all-cause mortality [59,60]. Notably, however, we found that sufficient dietary live microorganism intake significantly reduced all-cause and cardiovascular mortality in depressed individuals with sleep disorders, suggesting that dietary microorganisms could serve as an adjunctive therapy for depression [61] and improve long-term prognosis for this population.

To reduce the health hazards associated with sleep disorders, we recommend incorporating a variety of fermented and probiotic-rich foods into the diet, which can be categorized by their primary food sources and fermentation characteristics including fermented grains like sourdough bread, fermented rice porridge, and traditional rye kvass; fermented vegetables such as miso soup, fermented beetroot juice, kimchi, and pickled olives; fermented seafood like fermented fish and shrimp paste; dairy-based options such as yogurt, kefir, and unpasteurized cheese; dairy-free alternatives like coconut kefir, cashew-based probiotic blends, and kombucha; and ancient and specialty ferments like pineapple tepache, fermented cacao products, and raw honey (naturally containing beneficial microbes), with these foods collectively containing diverse live dietary microbes that may support sleep quality and long-term health by influencing gut microbiota or other biological pathways [62–64].

We believe that this study demonstrates notable innovation, as it is the first to explore the combined impact of dietary live microbes and sleep disorders on mortality risk. Our findings may offer new clinical guidance for populations with sleep disturbances. However, certain limitations should be acknowledged. First, the classification of dietary live microbes was based on expert consensus, which, though widely cited, may introduce inaccuracies due to the lack of precise microbial quantification in foods. Secondly, the use of self-reports to assess the situation of dietary intake and sleep disorders may not be able to capture the long-term dietary and sleep conditions, resulting in some unnecessary errors. Additionally, while we adjusted for numerous covariates, we may have been unable to include all considered variables that could influence the outcomes, such as food types, polyphenols in foods, and others. Furthermore, residual confounding from unmeasured factors also (such as medication use and preexisting gut diseases) could influence the results. Finally, the proposed

mechanisms (metabolic improvement and gut-brain axis modulation) remain speculative without experimental validation, limiting mechanistic interpretations. Despite these limitations, the study's conclusions retain their innovative and clinical relevance.

In summary, this study investigated the relationship between dietary live microbe intake, sleep status, and long-term health outcomes. Our analysis revealed that the mortality risks associated with low dietary microbe intake and sleep disorders exhibit a cumulative additive effect. Although the combined effect of high dietary microbes and sleep disorders showed only a trend toward reduced mortality risk, the observed associations between different exposure groups and mortality support the conclusion that dietary microbes may mitigate mortality risk in individuals with sleep disturbances. These findings suggest potential benefits of dietary microbe for long-term outcomes in this population, though further evidence is needed to confirm the relationship between dietary microbes, sleep, and mortality risk.

## 5. Conclusion

This study demonstrates that low dietary microorganism intake, when combined with sleep disorders, is associated with higher all-cause and cardiovascular mortality rates, whereas higher dietary microorganism intake correlates with improved long-term prognosis. These findings carry significant public health relevance, indicating that dietary microorganism intake may be highlighted in health and lifestyle recommendations to promote population health and survival outcomes.

## Supporting information

**S1 Table. The relationship between sleep quality, dietary microorganism intake, and ALL-cause mortality risk across diverse populations.**
(XLSX)

**S2 Table. The relationship between sleep quality, dietary microorganism intake, and CVD-cause mortality risk across diverse populations.**
(XLSX)

**S3 Table. Original data.**
(CSV)

**S1 File. Original R code.**
(DOCX)

## Author contributions

**Conceptualization:** Jun Li.

**Data curation:** Jun Li.

**Funding acquisition:** Jun Li.

**Software:** Jun Li, Tiantian Xue.

**Supervision:** Jun Li, Tiantian Xue.

**Writing – original draft:** XuanChun Huang, Lanshuo Hu, Ruikang Liu, Tiantian Xue.

**Writing – review & editing:** Jun Li.

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
