## [Decision Letter · Decision Letter 0]

Dear Dr. Li,

We look forward to receiving your revised manuscript.

Kind regards,

Seo Ah Hong, PhD

Academic Editor

PLOS ONE

2. Thank you for stating the following financial disclosure:  [This study was supported by National Key Research and Development Program of China (No. 2022YFC3500102), the National Natural Science Foundation of China (No. 81973836), the Beijing Municipal Science and Technology Development Funding Program of Traditional Chinese Medicine (No. JJ-2020-69), and High Level Chinese Medical Hospital Promotion Project (No. HLCMHPP2023065).].  Please state what role the funders took in the study.  If the funders had no role, please state: "The funders had no role in study design, data collection and analysis, decision to publish, or preparation of the manuscript." If this statement is not correct you must amend it as needed. Please include this amended Role of Funder statement in your cover letter; we will change the online submission form on your behalf.

Additional Editor Comments (if provided):

Reviewers' comments:

Reviewer's Responses to Questions

**Comments to the Author**

1. Is the manuscript technically sound, and do the data support the conclusions?

Reviewer #1: Yes

Reviewer #2: Partly

2. Has the statistical analysis been performed appropriately and rigorously?

Reviewer #1: Yes

Reviewer #2: No

3. Have the authors made all data underlying the findings in their manuscript fully available?

Reviewer #1: Yes

Reviewer #2: Yes

4. Is the manuscript presented in an intelligible fashion and written in standard English?

Reviewer #1: Yes

Reviewer #2: Yes

Reviewer #1: Important findings with a large sample on a fundamental topic in everyone's daily life like nutrition. Could you please clarify in the methods why subjects under the age of 20 were included and not 18 as expected? Also in the methods, the assessment of sleep disturbance in the subjects was done by a self-reported questionnaire. This assessment should be done by an official questionnaire like the Pittsburgh Sleep Quality Index. In introduction, the relationship between sleep disorders like OSA, bad dietary habits and brain disorders could be improved by the additional references like: Evaluation of wakefulness electroencephalogram in OSA patients. doi: 10.1007/s11325-024-03116-y.

Reviewer #2: - The study relies on expert consensus to classify dietary microbial content into broad categories (low, moderate, high) without laboratory-based quantification. This introduces potential inaccuracies in the measurement of actual microbial intake. Such imprecision could lead to misclassification bias, weakening the association between dietary microbial intake and mortality risk.

- The study does not control for medications (e.g., antibiotics, probiotics, or sleep aids) or pre-existing gastrointestinal conditions, both of which can significantly alter gut microbiota and influence outcomes.

The omission of these factors may confound the relationship between dietary microbes, sleep disorders, and mortality.

- The use of self-reported data for sleep disorders may introduce recall bias or inaccuracies. Additionally, sleep data is only assessed at baseline, without follow-up to track changes.

Subjective sleep data may not accurately capture the severity or nature of sleep disorders, leading to under- or overestimation of their association with mortality.

- Dietary data is based on a single 24-hour recall, which may not reflect habitual intake. This short-term dietary snapshot is used to draw conclusions about long-term health outcomes.Dietary habits can fluctuate, and a single-day recall may not accurately represent an individual’s typical intake of live microorganisms

- While the manuscript suggests mechanisms (e.g., gut-brain axis, neurotransmitter regulation), it does not experimentally validate these pathways. Proposed mechanisms remain speculative without direct evidence, limiting the study’s impact in advancing mechanistic understanding.

- The study does not account for lifestyle factors like stress, overall diet quality, or social support, which are known to affect sleep and mortality. These unmeasured factors may confound the observed relationships and obscure the role of dietary microorganisms.

**Do you want your identity to be public for this peer review?** For information about this choice, including consent withdrawal, please see our Privacy Policy

Reviewer #1: **Yes: ** Daniel Alfaiate

Reviewer #2: No

---

## [Author Response · Author response to Decision Letter 1]

29 Mar 2025

We would like to sincerely thank you for your time and effort in handling our manuscript. We deeply appreciate the thoughtful comments and valuable suggestions provided by both you and the reviewers, which have been instrumental in helping us improve our work. We are truly grateful for the opportunity to revise our manuscript under your guidance.

In response to the reviewers’ feedback, we have carefully revised the manuscript with a strong focus on enhancing clarity, improving logical flow, and refining our explanation of the study design. We have made every effort to address all the comments and have incorporated the necessary modifications in the relevant sections. We deeply appreciate this opportunity to improve our work, and we believe these revisions have strengthened the quality and clarity of our study. Specifically, we have made several major revisions. Firstly, we performed sensitivity analyses, which mainly included adjustments for additional variables such as antibiotics, probiotics, and social support. Secondly, in the Discussion section, we expanded the limitations subsection and added dietary recommendations for nutrient-rich foods.

Once again, we sincerely thank you for your time, effort, and invaluable guidance throughout this process. We are truly grateful for the opportunity to improve our manuscript and look forward to any further feedback you may have.

---

## [Decision Letter · Decision Letter 1]

Dear Dr. Li,

We look forward to receiving your revised manuscript.

Kind regards,

Assoc. Prof. Phakkharawat Sittiprapaporn, Ph.D.

Academic Editor

PLOS ONE

Reviewers' comments:

Reviewer's Responses to Questions

**Comments to the Author**

Reviewer #1: All comments have been addressed

Reviewer #2: All comments have been addressed

Reviewer #3: (No Response)

2. Is the manuscript technically sound, and do the data support the conclusions?

Reviewer #1: Yes

Reviewer #2: Yes

Reviewer #3: No

3. Has the statistical analysis been performed appropriately and rigorously?

Reviewer #1: Yes

Reviewer #2: Yes

Reviewer #3: No

4. Have the authors made all data underlying the findings in their manuscript fully available?

Reviewer #1: Yes

Reviewer #2: Yes

Reviewer #3: No

5. Is the manuscript presented in an intelligible fashion and written in standard English?

Reviewer #1: Yes

Reviewer #2: Yes

Reviewer #3: Yes

Reviewer #1: Thank you for the opportunity of reviewing this manuscript. Authors have addressed all suggestions properly. No further comments at this point.

Reviewer #2: Review Comments to the Author

Manuscript Title: Dietary Microbe Intake Might Mitigate Mortality Risk Exacerbated by Sleep Disorders: Evidence from NHANES

Manuscript Number: PONE-D-24-53480R1

The authors have made significant and commendable revisions to address the concerns raised in the prior round of peer review. Each reviewer comment has been carefully considered, and the responses provided are clear, well-reasoned, and backed by additional analyses or justified limitations. The incorporation of new sensitivity analyses (including antibiotics, probiotics, sleep aids, and social determinants of health) demonstrates a strong commitment to rigor and transparency. Furthermore, the manuscript now includes explicit acknowledgments of inherent limitations in sleep disorder classification and dietary assessment using NHANES data, which is appreciated.

Technical Soundness:

The manuscript presents a technically sound observational study using a nationally representative U.S. sample from NHANES (2005–2014) linked to mortality outcomes. The research question is timely and addresses the potential of dietary live microbes to modify the adverse health effects of sleep disorders. The data support the conclusions appropriately. The large sample size and the use of established statistical models (Kaplan-Meier and Cox regression) add credibility to the findings. The authors refrain from overstating causality and provide a balanced discussion, appropriately situating their findings within the current literature.

Statistical Analysis:

The statistical methodology is robust and well-executed. The three-tiered modeling approach with progressive adjustments is appropriate. The inclusion of subgroup and sensitivity analyses further strengthens the validity of the results. While formal model diagnostics are not described in detail, the statistical presentation is otherwise comprehensive and consistent with best practices.

Data Availability:

All data used are from NHANES, a public resource. The authors have provided a clear data availability statement and have included relevant tables and supplementary material. There is no concern about data transparency or compliance with PLOS’s open data policy.

Presentation and Language:

The manuscript is clearly written in standard English and follows a logical structure. The abstract and main text are coherent and intelligible. A minor issue is the title: “might mitigates” should be corrected to “might mitigate.” A few other minor grammatical issues (e.g., repetitive sentence openings in the response letter) could be polished, but overall, the language is of publishable quality.

Conclusion:

This is a well-conducted and meaningful epidemiological study that explores an under-investigated area of public health. The revised manuscript is stronger in both clarity and methodological rigor. The authors have adequately addressed all prior comments, and I find no major concerns remaining.

Recommendation:

Accept for publication. No further revisions are required.

Reviewer #3: The authors investigate an intriguing potential relationship between dietary live microbe intake and sleep disorder-related mortality. I appreciate the methods and analysis conducted, particularly the work taken to quantify live microbe intake. However, I have some questions and comments on the methodology and wording, outlined below:

1. The main flaw I see with the study design is the lack of consideration for the intake of other types of food as potential confounders. For example, there is literature on how an increased intake of dietary fibre (PMID: 38011755), polyphenols (PMID: 37290979), and resistant starch (PMID: 36570138), is associated with decreased mortality. Additionally, a decreased intake of foods such as ultra-processed food (PMID: 35231930) is associated with increased mortality. There could be a confounding relationship between the intake of these types of foods. For example, it could be reasonable to hypothesise that people who were more likely to eat microbe-rich foods would also be more likely to eat a diet high in fibre, or high in polyphenols and resistant starch. Are they also more likely to eat a diet low in processed food? This is particularly important when we consider the complex interaction between live dietary microbes and gut health. Literature suggests that the persistence and ameliorating effects of live microbe through diet are aided with intake of dietary fibre, i.e. “prebiotics” (PMID: 36276830). I feel that looking solely at dietary live microbes without considering other dietary factors is a narrow approach.

2. Regarding covariates in Table 1:

a. Why were only 3 “race” options included? There are other variables such as non-Hispanic Asian and Hispanic in the NHANES dataset.

b. Please be more specific with the schooling level variables. Does “>high school” mean some college and higher?

c. For smoking, I see that it was defined as “smoking history was defined as having smoked more than 100 cigarettes”, is that in the past year, lifetime?

d. And the drinking history “…was determined by whether the individual had

157 consumed at least one 12-ounce beer, 5-ounce glass of wine, or 1.5-ounce shot of liquor in the past year.” This categorisation does not seem sensitive enough and should be refined into more appropriate categories. For example, someone who consumes only 1-2 glasses of wine in a year should not be in the same category as someone who drinks wine daily.

3. In the results and conclusion of the abstract, the use of “risk” implies a causal link between dietary microbes and mortality which was not investigated in this study. The use of strong, causal language is misleading. I would revise to something implying an association only, as that is what was assessed.

4. Similarly, in the conclusion of the manuscript (lines 444-447), the sentence “This study demonstrates that a diet low in microorganisms, coupled with sleep disorders, can elevate all-cause and cardiovascular mortality risk, whereas increasing dietary microorganism intake may improve long-term prognosis, particularly reducing mortality risk in those with sleep disorders,” should be rephrased to imply that only an association was found, not a causal relationship.

5. The R code associated with analysis should be made available to keep in line with the Data Availability policy and for reproducibility.

6. The title seems to have a grammatical error, as “might mitigates” is incorrect. I would suggest revising to something more concise.

7. In the introduction (lines 80-82), in the statement, “Notably, recent evidence suggests that timely intake of dietary microorganisms may improve sleep by rebalancing gut microbiota[30],” the reference is for a study that was conducted on rats. This sentence should either be rephrased with that context or referenced with more appropriate literature.

8. Again, in the introduction (lines 354-355), the statement “Most studies focus on the impact of probiotics on sleep or disease outcomes” requires references.

**Do you want your identity to be public for this peer review?** For information about this choice, including consent withdrawal, please see our Privacy Policy

Reviewer #1: **Yes: ** Daniel Alfaiate

Reviewer #2: **Yes: ** Tien Hoang Anh

Reviewer #3: No

---

## [Author Response · Author response to Decision Letter 2]

29 May 2025

In response to the reviewers’ feedback, we have carefully revised the manuscript with a strong focus on enhancing clarity, improving logical flow, and refining our explanation of the study design. Specifically, we have made several major revisions: we have added adjustments for dietary fiber intake and processed food intake to our original sensitivity analysis. Additionally, in the abstract and conclusion sections, we have revised the wording to be more rigorous as per the reviewers' suggestions. Furthermore, we have uploaded our R code to facilitate reproducibility.

---

## [Decision Letter · Decision Letter 2]

Associations Between Dietary Microbe Intake and Mortality Risk in Individuals with Sleep Disorders: Evidence from NHANES

PONE-D-24-53480R2

Dear Dr. Li,

We’re pleased to inform you that your manuscript has been judged scientifically suitable for publication and will be formally accepted for publication once it meets all outstanding technical requirements.

Kind regards,

Assoc. Prof. Phakkharawat Sittiprapaporn, Ph.D.

Academic Editor

PLOS ONE

Additional Editor Comments (optional):

Reviewers' comments:

Reviewer's Responses to Questions

**Comments to the Author**

Reviewer #1: All comments have been addressed

Reviewer #3: All comments have been addressed

2. Is the manuscript technically sound, and do the data support the conclusions?

Reviewer #1: Yes

Reviewer #3: Yes

3. Has the statistical analysis been performed appropriately and rigorously?

Reviewer #1: Yes

Reviewer #3: Yes

4. Have the authors made all data underlying the findings in their manuscript fully available?

Reviewer #1: Yes

Reviewer #3: Yes

5. Is the manuscript presented in an intelligible fashion and written in standard English?

Reviewer #1: Yes

Reviewer #3: Yes

Reviewer #1: The authors have thoroughly addressed all of the questions and concerns raised by the reviewers in a clear and comprehensive manner. Each point has been thoughtfully considered, and appropriate revisions or explanations have been provided to ensure clarity, accuracy, and alignment with the reviewers’ feedback. The responses demonstrate the authors’ commitment to improving the quality of the manuscript and their attention to detail in responding constructively. As a result of these revisions and clarifications, all previously raised issues appear to have been resolved satisfactorily. At this stage, I have no further comments or suggestions for improvement.

Reviewer #3: Thank you to the authors for addressing all comments sufficiently. I appreciate the adjusted sensitivity analysis with additional dietary covariates, and agree the results have strengthened your findings. I also commend your work in rewording the content away from causal language to more accurately reflect the content of the manuscript. Congratulations on your publication!

**Do you want your identity to be public for this peer review?** For information about this choice, including consent withdrawal, please see our Privacy Policy

Reviewer #1: **Yes: ** Daniel Alfaiate

Reviewer #3: **Yes: ** Gina L Guzzo

---

## [Editor Report · Acceptance letter]

PONE-D-24-53480R2

PLOS ONE

Dear Dr. Li,

I'm pleased to inform you that your manuscript has been deemed suitable for publication in PLOS ONE. Congratulations! Your manuscript is now being handed over to our production team.

Kind regards,

on behalf of

Dr. PLOS Manuscript Reassignment

Staff Editor

PLOS ONE